



# In-stream *Escherichia Coli* Modeling Using high-
# temporal-resolution data with deep learning and
# process-based models
*Ather Abbas[1], Sangsoo Baek[1], Norbert Silvera[2], Bounsamay Soulileuth[3], Yakov Pachepsky[4]*
*Olivier Ribolzi[5], Laurie Boithias[5†], Kyung Hwa Cho[1†]*
[1] School of Urban and Environmental Engineering, Ulsan National Institute of Science and
Technology, Ulsan 689-798, Republic of Korea
[2] Institute of Ecology and Environmental Sciences of Paris (iEES-Paris), Sorbonne Université,
Univ Paris Est Creteil, IRD, CNRS, INRA, Paris, France
[3] IRD, IEES-Paris UMR 242, c/o National Agriculture and Forestry Research Institute,
Vientiane, Lao PDR
[4] Environmental Microbial and Food Safety Laboratory, USDA-ARS, Beltsville, MD, USA
[5] Géosciences Environnement Toulouse, Université de Toulouse, CNRS, IRD, UPS, Toulouse,
France
[†] *Co-corresponding authors: Kyung Hwa Cho (khcho@unist.ac.kr), Laurie Boithias*
*(laurie.boithias@get.omp.eu)*



## Abstract

Contamination of surface waters through microbiological pollutants is a major concern to public health. Although long-term and high-frequency *E. coli* monitoring can help prevent diseases from fecal pathogenic microorganisms, this monitoring is time consuming and expensive. Process-driven models are an alternative method for determining fecal pathogenic microorganisms. However, process-based modeling still has limitations in improving the model accuracy because of the complex mechanistic relationships among hydrological and environmental variables. On the other hand, with the rise in data availability and computation power, the use of data-driven models is increasing. Therefore, in this study, we simulated the transport of *Escherichia coli* (*E. coli*) in a 0.6 km² tropical headwater catchment located in Lao PDR using a deep learning model and a process-based model. The deep learning model was built using the long short-term memory (LSTM) technique, whereas the process-based model was constructed using the Hydrological Simulation Program–FORTRAN (HSPF). First, we calibrated both models for surface as well as for subsurface flow. Then, we simulated the *E. coli* transport with 6 min time steps with both the HSPF and LSTM models. The LSTM provided accurate results for surface and subsurface flow, by showing 0.51 and 0.64 of Nash–Sutcliffe Efficiency (NSE), respectively, whereas the NSE values yielded by the HSPF were -0.7 and 0.59 for surface and subsurface flow. The simulated *E. coli* concentration from LSTM also improved, yielding an NSE of 0.35, whereas the HSPF showed an unacceptable performance, with an NSE value of -3.01. This is because of the limitation of HSPF in capturing the dynamics of *E. coli* with land-use change. The simulated *E. coli* concentration showed rise and drop patterns corresponding to annual changes in land use. This study shows the application of deep learning-based models as an efficient alternative to process-based models for *E. coil* fate and transport simulation at the catchment scale.





**Keywords**: hydrological modeling; neural networks; fecal contamination; tropical rivers; South-
East Asia; hydrograph separation

## 1 Introduction

Contamination of surface waters through microbiological pollutants is a major public
health concern (Bain et al., 2014). Worldwide, pathogens have a propensity to wreak havoc on
human health because of the diseases they cause, such as diarrhea, resulting in infant mortality. In
particular, developing countries are vulnerable to pathogen-related diseases due to the deficit of
sanitation facilities (Boithias et al., 2016). *Escherichia coli* (*E. coli*) has been frequently used as
an indicator of fecal bacteria because it is easy to culture and less dangerous than other pathogens
(Rochelle-Newall et al., 2015). Higher concentrations of *E. coli* in water tend to be linked to fecal
pathogenic microorganisms, which are harmful to human health. Although long-term and high-
frequency *E. coli* monitoring can help prevent waterborne diseases from fecal pathogenic
microorganisms, the monitoring of *E. coli* concentration is time consuming and expensive (Cho et
al., 2016; Frolich et al., 2017; Kim et al., 2017). High-frequency datasets of *E. coli* concentration
are scarce, and available long-term datasets are often inadequate to yield a continuous
concentration of fecal pathogenic microorganisms (van der Leeuw, 2004). This drawback in
monitoring can be overcome by modeling approaches. Thus, they can be an alternative to
determinin the fate and transport of fecal pathogenic microorganisms at the catchment scale by
simulating *E. coli* in each one of the environmental compartments, for example the soil surface
and streams (Ligaray et al., 2016; Perez-Pedini et al., 2005; Pacehpsky et al., 2011).



Several process-based models have been developed to model stream water contamination
by *E. coli*. Popular models to simulate *E. coli* are the Soil and Water Assessment Tool (SWAT)
(Neitsch et al., 2011), Hydrological Simulation Program–FORTRAN (HSPF) (Bicknell et al.,
1997), INCA-pathogen (Whitehead et al., 2016), and Pathogen Catchment Budget (PCB)
(Ferguson et al., 2007). The fate and transport of *E. coli* is a complex phenomenon that depends
on several drivers (Pachepsky et al., 2018), such as the hydrological regime (Boithias et al., 2016;
Pachepsky et al., 2017), relative contributions of both surface runoff and subsurface flow to the
overall in-stream discharge (Boithias et al., 2021), concentration and sources of suspended
sediment (Ribolzi et al., 2016; Nguyen et al., 2016), land use (Causse et al., 2015; Nakhle et al.,
2021), intrinsic properties of the bacterium (Pachepsky et al., 2014), and economic conditions
(Iqbal et al., 2019). However, the process-based model still has limitations in terms of high
accuracy due to complex mechanistic relationships among hydrological and environmental
variables (Abimbola et al., 2020). In addition, the simplified equations of these models might
increase the inherent uncertainties, resulting in simulation errors. The *E. coli* concentration in
surface water varies significantly within a very short span of time (Chen et al., 2014; Boithias et
al., 2021). Daily and weekly simulations cannot capture the dynamics of *E. coli* in a short duration.
In particular, the simulation with high-resolution frequency is important in small headwater
catchments because the duration of flood events might be less than one day (Gassman et al., 2007).
Therefore, an *E. coli* concentration simulation with high-frequency resolution should be conducted
to determine the temporal distribution of *E. coli*.

Recently, deep learning (DL) has become a promising alternative approach for estimating

water quality by using features of water constituent dynamics (Pyo et al., 2021). Long short-term
memory (LSTM) networks have an advantage over other deep learning-based models in that they



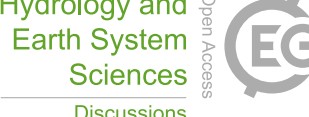

88 can extract complex patterns from sequence data (Schmidthuber and Hochreiter, 1997). Several

89 studies have applied deep learning to water quality modeling and prediction (Peterson et al., 2020;

90 Isikdogan et al., 2017; Solanki et al., 2015). Dong et al. (2019) used LSTM to predict dissolved

91 oxygen and showed that LSTM performs better than machine learning methods, such as

92 autoregressive integrated moving average or artificial neural networks. Although LSTM has been

93 used extensively for building hydrological models (Abbas et al., 2020), its potential has not yet

94 been explored to estimate *E. coli* concentration in stream waters. Deep learning-based models have

95 also not been developed for the simulation of water quality with high-resolution frequency.

96  This study aims to evaluate the applicability of LSTM to simulate in-stream *E. coli*

97 concentration with high temporal resolution. In addition, the process-based model HSPF was used

98 as a benchmark to compare and assess the performance of LSTM. Both models were applied in a

99 0.6 km² tropical headwater catchment from the northern Lao People's Democratic Republic (PDR).

100 The temporal resolution of the simulations was 6 min in both models. Thus, the specific objectives

101 of this study were to compare the performance of a process-based model and a deep learning model

102 1) to simulate both surface and subsurface flow, 2) to simulate *E. coli* concentration, and 3) to

103 analyze the response of *E. coli* by changing land use.



## 2 Materials and Methods

### 2.1 Study site and data acquisition

The study area is the Houay Pano headwater catchment, located 10 km south of the city of Luang Prabang, Lao PDR (Boithias et al., 2021) (Fig. 1). This catchment is representative of a montane agroecosystem in Southeast Asia and is part of the long-term critical zone observatories' network called multiscale TROPIcal CatchmentS (M-TROPICS), which is affiliated with the French research infrastructure OZCAR (Gaillardet et al., 2018). This site had undergone rapid land-use changes from 2011 to 2018 (Fig. S1a). The characteristics of this area, including land use information, are provided in the supplementary information (Text S1). We collected climate, hydrological, *E. coli* concentration, and electrical conductivity data at 6 min time steps from 2011 to 2018. Rainfall, relative humidity, solar radiation, wind speed, and air temperature were measured with an automatic weather station Campbell Scientific BWS200, which was equipped with ARG100 (a 0.2 mm capacity tipping bucket). The potential evapotranspiration was calculated using the Penman–Monteith method. We measured the stream water level at the monitoring station using a V-notch and water-level recorder (OTT Thalimedes). The discharge was estimated based on the rating curve between the discharge and water levels. The surface and subsurface flow were calculated using the electrical conductivity method (Ribolzi et al., 2018). A detailed description of this method is provided in the supplementary information (Text S2). *E. coli* concentration was measured based on the standardized microplate method (ISO 9308–3). A detailed explanation of *the E. coli* experiment can be found in the supplementary information (Text S3). In this study, we carried out biweekly grab sampling of *E. coli* from 2011 to 2018. Over the same period, we also specifically sampled 11 flood events to assess *E. coli* dynamics during flood events by using an



automated sampler (ICRISAT) triggered by the water level recorder to collect water after every 2
cm water level change during flood rising and every 5 cm water level change during flood
recession. The total number of *E. coli* samples collected over the 2011–2018 period was 255. In
addition, we collected the monthly number of poultries, swine, goats, and the number of humans
who visited the study area. These data were used to quantify the source of *E. coli* in this catchment
(Rochelle-Newall et al., 2016) (Fig. S1b).

**2.2 Flow and *E. coli* concentration simulation.**

In this study, HSPF and LSTM models were used to simulate in-stream surface flow,

subsurface flow, and *E. coli* concentration. HSPF and LSTM are popular models among the
process-based and DL models (Bicknell et al., 1997; Ahmadisharaf and Benham, 2020; Kratzert
et al., 2019). Both models have been adopted for hydrological and water quality simulations
(Peterson et al., 2020; Isikdogan et al., 2017; Ahmed et al., 2014). In the HSPF, the simulation of
surface and subsurface flow and of *E. coli* concentration was carried out in three steps: (1) building
the model, (2) conducting sensitivity analysis based on the Latin-Hypercube–One-factor-At-a-
Time (LH-OAT), and 3) calibrating the model using the Newton algorithm (Nash, 1984). A
schematic of the LSTM simulation is shown in Fig. 2. The first step in building this model was
data preparation (Fig. 2a). LSTM then simulated surface and subsurface flow with climate data
(Fig. 2b). Finally, we estimated the *E. coli* concentration at 6 min intervals using rainfall, bacteria
source, land-use change, and surface and subsurface flow (Fig. 2c). Both models considered the
source of *E. coli* to simulate its concentration at the catchment outlet. The fecal matter from the *E.*
*coli* sources was assumed to be evenly distributed in the catchment. The monthly *E. coli* source





data is presented in Fig. S1b. The time series data of the *E. coli* source was used as input for the *E.*
*coil* simulation.



### 2.2.1 Hydrological Simulation Program Fortran (HSPF)

The HPSF model is a process-driven model that simulates processes at the catchment scale (Bicknell et al., 1997). It has been extensively used to model the fate and transport of *E. coli* in catchments (Ahmadisharaf and Benham, 2020; Chin et al., 2009) and to develop total maximum daily loads of *E. coli* at various locations (Mishra et al. 2018; Yagow et al., 1998). The original software was written in the FORTRAN programming language. Recently, the Hydrological Simulation Program Python (HSP2) was developed based on the Python programming language (van Rossum, 2007). HSP2 is a platform-independent software that extends the functionality of HSPF by allowing the use of dynamic variables and easier management of input and output files (Heaphy et al., 2015). The HSPF simulates the hydrological cycle by discretizing the catchment into pervious and impervious hydrological response units (HRUs). Previous HRUs simulate evapotranspiration, surface detention, surface infiltration, interflow, baseflow, and deep percolation, whereas impervious HRUs simulate surface detention and surface flow (Bicknell et al., 1997). The simulation of in-stream *E. coli* concentration in HSPF is based on a first-order kinetics approach, considering the decay rate (Fonseca et al., 2014). Detailed descriptions of hydrological and *E. coli* simulations can be found in Bicknell et al. (1997). For this study, we rewrote the modules of *E. coli* simulation, and the simulation was carried out in the Python programming language. This allowed us to incorporate more dynamic use of input data, such as the annual change in land use and the monthly bacterial source.

In our study, HRUs were divided into four units based on land use: Forest, Fallow, Teak, and Annual crop. Among land uses, we did not consider any imperviousness in the Forest and Fallow. We considered 2 % and 1 % imperviousness for the Teak and Annual crop land uses (Patin





et al., 2018). We selected 13 and 4 parameters for each land use for the sensitivity analysis of
hydrological and *E. coil* simulations, respectively (Table 1 and Table S1). The total number of
parameters for hydrological and *E. coil* simulation were 52 and 18, respectively. In model
calibration, we selected the 25 most sensitive parameters of the hydrological simulation and all
parameters of the *E. coli* simulation. Sensitivity analysis and model calibration were conducted
based on the LH-OAT and the Newton algorithm, respectively. A detailed explanation of the LH-
OAT and the Newton algorithm can be found in the Supplementary Information (Text S4).





### 2.2.2 Long short-term memory (LSTM)


In the data preparation step (Fig. 2a), our data were converted to a 6 min frequency. We
then built the LSTM model to simulate surface and subsurface flow using the validated model
structure (Abbas et al., 2020) (Fig. 2b). It uses historical data of rainfall, solar radiation, air
temperature, and potential evapotranspiration to simulate surface and subsurface flow. To simulate
the output at a time-step "t," LSTM uses the data of previous "n" time steps as inputs (Chollet,
2018). The inputs from previous time steps are used by LSTM to predict the output at the next
time step (t+1). The number of these time steps "n" are called lookback steps (Chollet, 2018).  The
simulated surface and subsurface flow from the LSTM were applied to simulate the *E. coli*
concentration (Fig. 2c). We adopted a bacterial source and land-use information as input for the
LSTM. To investigate the impact of land-use change on in-stream *E. coli* concentration, we
conducted *E. coli* simulations in two scenarios. In scenario 1, we used the land-use change and *E.*
*coli* source information separately. In scenario 2, we calculated the *E. coli* source per area for each
land use.
LSTM is a special type of recurrent neural network designed to extract temporal features
from sequence data (Hochreiter and Schmidhuber, 1997). An LSTM cell is the basic building block
of the LSTM (Fig. S2). It consists of three "gates" and two "states" The gates are "forget," "update,"
and "output," which decide what information to forget, allow in, and allow out from the LSTM
"memory," respectively. The states act as a memory or information carrier across time. The
equations describing the functions of gates and states are as follows:

$$C_c^{<t>} = tanh(W_c[\,h^{<t-1>}, x^{<t>}] + b_c), \qquad (1)$$

$$\Gamma_f = \sigma(W_f[c^{<t-1>}, x^{<t>}] + b_{f)}, \qquad (2)$$



$$\Gamma_o = \sigma(W_o[c^{<t-1>}, x^{<t>}] + b_{o),} \tag{3}$$

$$\Gamma_u = \sigma(W_u[c^{<t-1>}, x^{<t>}] + b_{u),} \tag{4}$$

$$C^{<t>} = \Gamma_u * C_c^{<t>} + \Gamma_f * C^{<t-1>} \tag{5}$$

$$h^{<t>} = \Gamma_o * tanh\ C^{<t>}. \tag{6}$$


The symbol $*$ in the above equations represents elementwise multiplication. The behavior

of each gate is controlled by the weights ($W$) and biases ($b$) associated with them. Their output
was further modified by a nonlinear function ($\sigma$). At each time step ($t$), the prospective cell state
($C_c^{<t>}$) is calculated based on the output from the previous time step ($h^{<t-1>}$) and the input from
the current time step ($x^{<t>}$) (Eq. 1). The notation $W_c[\,h^{<t-1>}, x^{<t>}]$ represents pointwise
multiplication of new inputs and previous hidden state with the weight matrix $W_c$ separately and
then adding their output. This prospective cell state ($C_c^{<t>}$), along with the output from the "forget"
and "update" gate decides the current cell state ($c^{<t>}$) (Eq. 5). The current cell state and output
gate control the output values from LSTM ($h^{<t>}$), the so-called hidden state (Eq. 6). The
hyperbolic tangent ($tanh$) is another nonlinearity used in LSTM for the calculation of the cell state
(Eq. 1) and the output state (Eq. 6). Equations 1–6 are used to calculate the LSTM output, which
is then compared with observed values to calculate the error. This study used the mean square error
(MSE) as the error function.

We used the TensorFlow software v1.15 for building the LSTM model (Abadi et al., 2016).

We used an Intel® Core™ i7-9700 processor with a graphics card of NVIDIA GeForce RTX 2080
having 12 GB of dedicated GPU memory, along with 64 GB of Random-Access Memory for
simulating surface, subsurface, and *E. coli.*






### 2.2.3 Hyperparameters of LSTM


The structure and performance of the LSTM were controlled by hyperparameters,

including the dropout rate, LSTM units, learning rate, lookback steps, and activation functions for
both LSTM and the fully connected layer (Table 2). Dropout is a regularization technique that
switches off a certain number of nodes in the LSTM (Goodfellow et al., 2016). This simple
technique helps break the brittle coadaptation of weights, which hinders generalization to unseen
data. This way, dropout prevents overfitting (Srivastava et al., 2014). In overfitting, the model
performs better on calibration data, but its performance deteriorates on new unseen data. The
number of LSTM units directly corresponds to the learning capacity of LSTM, but it also accounts
for more memory and computation. This number determines the size of the weight matrix of an
LSTM. The learning rate defines the change in the weights of the neural network during calibration
(Goodfellow et al., 2016). A higher number of lookback steps allows LSTM to capture long-term
patterns at the cost of an increase in memory consumption and computation. The activation
function determines the nonlinearity in the model.

### 2.3 Performance statistics


Evaluations to assess the performance of the HSPF and LSTM were conducted

using MSE, Nash–Sutcliffe efficiency (NSE), and percent bias (PBIAS) (Nash and Sutcliffe,
1970; Gupta et al., 1999). NSE is useful for interpreting the model performance by generating a
dimensionless value as the performance index (Lin et al., 2017). The PBIAS measures the
average tendency of the simulated data to be overestimated or underestimated than observed





values (Moriasi 2007). The MSE, NSE, and PBIAS were calculated using the following
equations:

$$MSE = \frac{[\sum_{i=1}^{n}(o_i - p_i)^2]}{n} \tag{7}$$

$$NSE = 1 - \frac{\sum(o_i - p_i)^2}{\sum(o_i - \bar{o})^2} \tag{8}$$

$$PBIAS = 1 - \frac{\sum_{i=1}^{n} o_i - p_i}{\sum_{i=1}^{n} o_i} \tag{9}$$

where $p_i$ is the simulated data, $o_i$ is the observed data, and $n$ is the number of points in the data.



## 3 Results and discussion

### 3.1 Land use change and *E. coli* source

The land-use change from 2011 to 2018 is shown in Fig. S1a. The area of Fallow land-use increased from 2011 to 2016, whereas Annual crop area decreased. Teak tree plantations were expanded until 2013 and were retained. Forest land use accounted for about 10 % of the study area from 2011 to 2018. In general, the land-use change has been dynamic from 2011 to 2013, whereas its variation diminished from 2016 to 2018. Previous studies have demonstrated that the expansion of Teak trees might increase the surface flow (Ribolzi et al., 2017; Song et al., 2020). Higher runoff at the soil surface may cause a higher inflow of *E. coli* with surface flow. The monthly *E. coli* source in the catchment decreased from $2 \times 10^{15}$ in 2011 to $3 \times 10^{14}$ in 2018 (Fig. S1b). This decrease in *E. coli* source is caused by the decrease in manpower needed in Teak tree plantations and in Fallow plots, compared to the Annual crop (Fig. S1a) (Boithias et al., 2021).





### 3.2 Sensitivity analysis and optimization result


The sensitivity results for the flow simulation are shown in Fig. S3, and the most sensitive
parameters are listed in Table S2. The interflow and infiltration-related parameters were the most
sensitive parameters for surface and subsurface flows. The Manning's "n" value (NSUR) for Teak
and Fallow land uses was among the 10 most sensitive parameters. Kim et al. (2017) suggested
that Manning's value is the most sensitive parameter in the hydrological simulation of tropical
headwater catchments, such as the Houay Pano catchment in northern Lao PDR. The groundwater
recession rate (AGWRC) and soil infiltration capacity (INFILD) were sensitive to subsurface flow.
In Annual crop land use, infiltration capacity (INFILT) and upper zone storage (UZSN) were the
most sensitive parameters. Abbas et al. (2020) demonstrated that INFILT is the most sensitive
parameter for subsurface flow in tropical subcatchments.
The sensitivity analysis results for *E. coli* are shown in Fig. S4 and Table S3. The
parameters related to the transport of *E. coli* on the land surface (e.g., WSQOP, SQOLIM_MF)
were more sensitive than other parameters. IOQC and AOQC were the least sensitive parameters.
These parameters are related to *E. coli* transport in interflow and baseflow (Bicknell et al., 2011).
This implies that the in-stream *E. coli* concentration at the study site is mainly driven by surface
flow (Boithias et al., 2021). A previous study also demonstrated that 89 % of in-stream *E. coli*
concentrations were driven by surface flow (Boithias et al., 2021). Figure 3 shows the model
performance dependent on different objective functions. We found that the model performance
was better when the NSE was selected as the objective function. The NSE of the surface and
subsurface flow was positive by optimizing with NSE. However, the NSE value for surface flow





was negative when the objective function was MSE during the optimization. Negative NSE
indicated an "unsatisfactory" performance range (Moriasi et al., 2015).

**3.3 Flow simulation**

The simulated surface and subsurface flow using the HSPF are plotted in Fig. 4. We found

that the simulated subsurface flow was underestimated compared to the observations. Although
surface flow from the HSPF followed the trend and peaks of observations, this model yielded a
negative NSE value, indicating that the model simulation was unacceptable (Moriasi et al., 2015)
(Table 3). The NSE values for subsurface flow from HSPF were 0.49 and 0.59 for calibration and
validation, respectively. Hence, the HSPF model is better at simulating subsurface flow than
surface flow. In particular, the simulated surface flow was underestimated compared to the
observations. The average values of INFILT and UZSN were 0.36 and 1.22, respectively, which
were larger than those reported in previous studies (Lee et al., 2020). INIFILT controls the overall
division of available moisture into the surface and subsurface (Bicknell et al., 2001). The parameter
UZSN influences the evapotranspiration process (Bicknell et al., 2001). This underestimation of
surface flow using HSPF is consistent with a previous study (Kim et al., 2017). We also
investigated the impact of underestimation and overestimation of the flow by plotting flow
duration curves (Fig. S5). Although both flows can capture the peak flow, the simulated subsurface
flow was still underestimated compared to the observed subsurface flow.

The simulated surface and subsurface flows using the LSTM model are plotted in Fig. 5.

The NSE values for the calibration period were 0.56 and 0.69 for surface and subsurface flow,
respectively. The corresponding validation NSE of the surface and subsurface flow were 0.51 and



0.64, respectively. These results indicate that the LSTM had a satisfactory performance for both
the calibration and validation periods according to the criteria of Moriasi et al. (2015). LSTM
overcame the problem of the HSPF model underestimating subsurface flow. In addition, the peak
surface flows from the LSTM were similar to observations. The observed and simulated flows in
storm events are presented in Figs. S6–S11. LSTM can follow the observed trends in surface and
subsurface flow more closely than the HSPF. This leads to increased NSE values for both surface
flow as well as for subsurface flow. The hyperparameters of the LSTM are described in Table 2.
The rectified linear unit (ReLU) was chosen as the activation function for the LSTM output.
Because the simulated *E. coli* should be positive, we chose ReLU, which cannot produce negative
values from the model (Nair and Hinton, 2010). The optimal batch size and LSTM units were 16
and 100, respectively. The optimal value of the lookback steps was 50, which is equal to 5 h of
input data.

We analyzed the model performance for surface and subsurface flows during storm events

(Fig. 6). These events were selected where the peak flow exceeded 0.2 m per s. The performance
of LSTM is considerably better than that of HSPF for most storm events. In surface flow, the
average MSE of LSTM and HSPF was 1.1e-4 and 6.1e-4 ($m^3 s^{-1}$), respectively. The NSE values
from LSTM varied from 0.2 to 0.6, whereas that of HSPF ranged from -1.0 to 0.4. We found that
the NSE values from the HSPF vary considerably depending on storm events. On June 11, 2015,
the NSE value of HSPF was as high as 0.4, whereas for some others it was below 0. Although the
subsurface flow of the HSPF provided better model performance than surface flow simulation, this
model still presented an unacceptable result with a negative NSE value.



### 3.4 *E. coli* simulation


Figure 7 shows the temporal distribution of *E. coli* concentration using HSPF and LSTM.
The *E. coli* concentration from HSPF was overestimated compared to the observed *E. coli*
concentration. The performance matrices of the HSPF were also worse than those of the LSTM
(Table 4). In particular, the HSPF simulation presented a PBIAS value of 73, indicating an
overestimation of *E. coli* concentration (Moriasi et al., 2015). Ackerman and Weisman (2014)
reported that the *E. coli* simulation from HPSF was overestimated compared to observation. The
overestimation of simulated *E. coli* at tropical sites has also been observed by Kim et al. (2017).
*E. coli* simulation from LSTM is satisfactory in both calibration and validation periods according
to the criteria set by Moriasi et al. (2015). In contrast, the HSPF result can be regarded as
"unsatisfactory" in both the calibration and validation periods. These results implied that LSTM
could generate acceptable performances and had good agreement between the observed and
simulated *E. coli*.
The simulation during the storm events using both the HSPF and LSTM models are
shown in Fig. 8 and Figs. S6–S11. Figure 8 shows the storm events from the validation data,
whereas the other figures show the storm events from the calibration data. In general, the simulated
*E. coli* by HPSF and LSTM were overestimated and underestimated, respectively. This difference
might be caused by the fact that *E. coli* from HSPF is more responsive to surface flow, wheras *E.*
*coli* from LSTM is more influenced by subsurface flow (Ackerman and Weisman, 2014). The
sensitivity analysis of HSPF also demonstrated that the influence of interflow and baseflow on *E.*
*coli* is weaker than surface flow because the parameters IOQC and AOQC are the least sensitive
parameters for *E. coli* simulation. Both parameters affect the *E. coli* concentration in interflow and





baseflow (Bicknell et al., 2001). The simulated *E. coli* of LSTM rose sharply and dropped slowly,
similar to the observations, whereas that of the HSPF decreased steeply (Figures S6–S11).
Although both models simulated the peak time of the *E. coil* correctly, the HSPF was limited to
simulate a slope in its falling limb. This performance difference between both models was caused
by the extent of influence from hydrological variables (e.g., rainfall, surface flow, and subsurface
flow) to model output. LSTM was effective in reflecting the response of variables to output
(Kratzert et al., 2019).

The performance matrices for the LSTM and HSPF models during storm events are shown

in Fig. 9. In general, we observed better LSTM performance than HSPF for both NSE and MSE
values. The HSPF model performed better than the LSTM for only two storm events on June 15,
2014, and June 11, 2015. For the remaining storm events, the NSE values from LSTM are higher
than those of the HSPF—an NSE range from 0.20 to 0.65. Similarly, for MSE values, the LSTM
was superior to the HSPF for all storm events except for the storm events of June 15, 2014, and
June 11, 2015.

We observed the impact of logarithmic and minmax transformations on model performance

(Fig. 10). The result of the logarithmic transformation was closer to the observation than the
minmax transformation by showing an NSE of 0.57. A negative PBIAS value was obtained in
logarithmic transformation. This indicated that the simulated *E. coli* from logarithmic
transformation was underestimated, whereas the result of the minmax transformation was
overestimated. The reason for this behavior can be attributed to the ability of minmax scaler to be
more sensitive to outliers (Chuang et al., 2010). As a result, if a better accuracy during storm events
is required, the target variable can be transformed on a logarithmic scale prior to calibration. This
is because log transformation can reduce the effect of outliers from data (Singh and Kingsbury,





2017). It has been reported that log transformation can improve the performance of data-driven
models when the data contain outliers (Zheng and Casari, 2018).

**3.5 *E. coli* response to land-use change**
We investigated the impact of land-use change and bacterial sources on the in-stream *E.*
*coli* concentration simulation (Fig. 11). In scenario 1, we used land-use change time-series
information (Fig. S2a) and bacterial source information (Fig. S2b). In scenario 2, we divided the
bacterial source by the fraction of each land use (Fig. S2c). In scenario 1, we observed a larger
variation in *E. coli* concentration from 2014 to 2018 (Fig. 11a), whereas in scenario 2, the variation
in *E. coli* was smaller than that in scenario 1 (Fig. 11b). This variation in *E. coli* was due to land-
use change in scenario 1. In particular, *E. coli* in 2016 was less than in other years because Annual
crop land use decreased. On the other hand, the variation in *E. coli* was not observed in scenario 2
from 2015 to 2017. Neither scenario showed a significant response from 2011 to 2014. During
these years, the rise in Fallow land use was complemented by a decrease in Annual crop land use.





### 3.6 Limitations and future research


Transport of soil particles by surface flow and suspended sediments within the stream play
a crucial role in the fate and transport of *E. coli* (Thupaki et al., 2013). Several studies have
emphasized the importance of particle size (Cho et al., 2010), adsorption to soil and sediment
particles (Palmateer et al., 1993), and resuspension of *E. coli* (Kim et al., 2017) with streambed
sediments for modeling the fate and transport of *E. coli* at the catchment scale. In this study, we
did not consider sediment transport, nor the attachment/detachment of *E. coli* on/from soil particles
and suspended sediments. Several studies have been conducted on the monitoring and modeling
of *E. coli* without considering sediment transport (Ahmadisharaf and Benham, 2020; Mishra et al.,
2018). However, the need for its inclusion has been indicated elsewhere (Pandey and Soupir, 2013).
To model sediment transport, additional data on suspended sediment concentration are required to
build both the HSPF and deep learning-based models. Therefore, this modeling exercise can be
further improved by collecting sediment-related data and modeling sediment transport along with
*E. coli* concentration.





## 4 Conclusions

In this study, we simulated the transport of bacteria in a headwater catchment of the northern

Lao PDR at 6 min time steps. The main findings of this study are summarized as follows:

- Both the LSTM and HSPF models can accommodate land-use change and bacteria-
  source variation with time.

- The performance of the surface and subsurface flow simulation of LSTM was superior for
  both the calibration and validation steps when compared with the HSPF. The LSTM
  provided accurate results for surface and subsurface flow by showing NSE values of 0.51
  and 0.59, respectively, whereas the HSPF showed -0.7 and 0.55 of NSE.

- Our LSTM model showed better performance compared to HSPF for *E. coli* simulation.
  The NSE of the HSPF and LSTM were -3.01 and 0.35, respectively. We found that the
  LSTM model can respond to changes in land use.

This study shows that deep learning-based models are an efficient alternative to process-based

models to simulate *E. coli* in a given catchment. Because LSTM can generate reasonable *E. coli*

simulations, it could be applied to provide effective strategies for diseases that wreak havoc on

human health. Therefore, a deep learning approach can be useful in developing better water

sustainability and management.





## Code Availability

Programming Language: Python

Software development: PyCharm

Year first available: 2021

Software Availability: contact the authors

Contact Address: School of Urban and Environmental Engineering, Ulsan National Institute of

Science and Technology, UNIST-gil 50, Ulsan, 689–798, Republic of Korea. E-mail:

khcho@unist.ac.kr

## Author contributions

**Ather Abbas:** Conceptualization, Data curation, Methodology, Visualization, Writing - original

draft, Writing - review & editing. **Sangsoo Baek:** Visualization, Writing - review & editing.

**Olivier Ribolzi:** review & editing. **Norbert Silvera:** Data curation. **Bounsamay Soulileuth**:

Sampling and data preparation. **Yakov Pachepsky:** review & editing. **Laurie Boithias:** Funding

acquisition, Supervision, Validation, Writing - review & editing. **Kyung Hwa Cho:**

Conceptualization, Funding acquisition, Supervision, Validation, Writing - review & editing.

## Competing Interests

The authors declare that they have no conflict of interest.





**Acknowledgments**
This study was supported by Basic Science Research Program through the National Research
Foundation of Korea (NRF) funded by the Ministry of education (No. 2017R1D1A1B04033074).
The authors sincerely thank the Lao Department of Agricultural Land Management (DALaM) for
its support, including granting the permission for field access, and the M-TROPICS Critical Zone
Observatory (https://mtropics.obs-mip.fr/), which belongs to the French Research Infrastructure
OZCAR (http://www.ozcar-ri.org/) for data access.



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





**Table 1:** Optimal values and range of HSPF parameters for surface and sub-surface flows and *E. coli* concentration. Bold parameters were optimized during flow calibration process. All parameters related to *E. coli* were optimized during model calibration.

|  | Parameters | Land use | | | | Lower Limit | Upper Limit |
|---|---|---|---|---|---|---|---|
|  |  | Forest | Teak | Fallow | Annual Crop |  |  |
| Surface and sub-surface flow | INFILT | **0.31** | **0.39** | **0.39** | **0.36** | 0.001 | 0.5 |
|  | INFILD | 2.0 | **1.94** | **1.95** | **1.55** | 1.0 | 3.0 |
|  | INTFW | 2.60 | **7.01** | **7.01** | 5.64 | 1.0 | 10.0 |
|  | UZSN | 1.36 | 1.47 | **0.84** | **1.24** | 0.05 | 2.0 |
|  | LZSN | **8.88** | **9.43** | **4.18** | **8.66** | 2.0 | 10.0 |
|  | AGWETP | 0.02 | **0.007** | 0.02 | **0.06** | 0.0 | 0.2 |
|  | NSUR | **0.18** | **0.39** | **0.15** | **0.43** | 0.05 | 0.5 |
|  | BASETP | **0.05** | **0.09** | 0.095 | **0.003** | 0.0 | 0.2 |
|  | DEEPFR | 0.28 | 0.16 | 0.21 | **0.20** | 0.0 | 0.5 |
| *E. coli* concentration | SQOLIM MF | 4.99 | 1.35 | 2.04 | 0.53 | 0.5 | 10 |
|  | WSQOP | 9.12 | 9.31 | 8.87 | 9.38 | 0.1 | 10.0 |
|  | IOQC | 5367 | 8337 | 8380 | 8756 | 1000 | 10000 |
|  | AOQC | 8672 | 7474 | 5465 | 8776 | 1000 | 10000 |
|  | FSTDEC |  | 3.04 |  |  | 0.1 | 10.0 |
|  | THFST |  | 1.92 |  |  | 1.01 | 2.0 |





**Table 2:** Hyper-parameters of LSTM for surface flow, sub-surface flow and *E. coli*

concentration simulation.

| Parameter | Surface and sub-surface flow | *E. coli* |
|---|---|---|
| **Activation function (LSTM Layer)** | Rectified Linear Unit (ReLU) | Rectified Linear Unit (ReLU) |
| **Activation function (Dense Layer)** | Rectified Linear Unit (ReLU) | Rectified Linear Unit (ReLU) |
| **Batch size** | 128 | 16 |
| **Learning rate** | 1e-5 | 1e-6 |
| **lookback steps** | 5 hours | 5 hours |
| **Dropout** | 0.3 | 0.3 |
| **Hidden units** | 64 | 100 |
| **Input data** | Rainfall, Solar Radiation, Air Temperature, Potential Evapotranspiration | Rainfall, Surface flow, Sub-surface flow, Land use, Bacteria source |
| **Calibration epochs** | 500 | 7000 |
| **Training samples** | 490000 | 182 |
| **Test samples** | 210000 | 73 |




**Table 3:** Performance metrics of HSPF and LSTM model for surface and sub-surface flow.

| Model | Flow Type | Scenario | MSE $(m^3 s^{-1})$ | NSE | PBIAS |
|---|---|---|---|---|---|
| HSPF | Surface Flow | Calibration | 6.4e-4 | -0.02 | -59 |
| | | Validation | 4.7e-5 | -0.7 | -28 |
| | Sub-surface Flow | Calibration | 2.7e-4 | 0.49 | -51 |
| | | Validation | 5.e-4 | 0.59 | -22 |
| LSTM | Surface Flow | Calibration | 1.4e-4 | 0.56 | -48 |
| | | Validation | 1.9e-4 | 0.51 | -63 |
| | Sub-surface Flow | Calibration | 5.4e-3 | 0.69 | -42 |
| | | Validation | 5.9e-3 | 0.64 | -46 |






**Table 4:** Performance metrics of HSPF and LSTM for *E. coli* concentration simulation.

| Model | Scenario | MSE (MPN 100 mL$^{-1}$) | NSE | PBIAS |
|-------|----------|-------------------------|-----|-------|
| **HSPF** | Calibration | $1.4e^8$ | -0.29 | -58 |
| | Validation | $1.9e^8$ | -3.01 | 73.01 |
| **LSTM** | Calibration | $7.1e^6$ | 0.39 | -1.49 |
| | Validation | $3.0e^7$ | 0.35 | 62.72 |




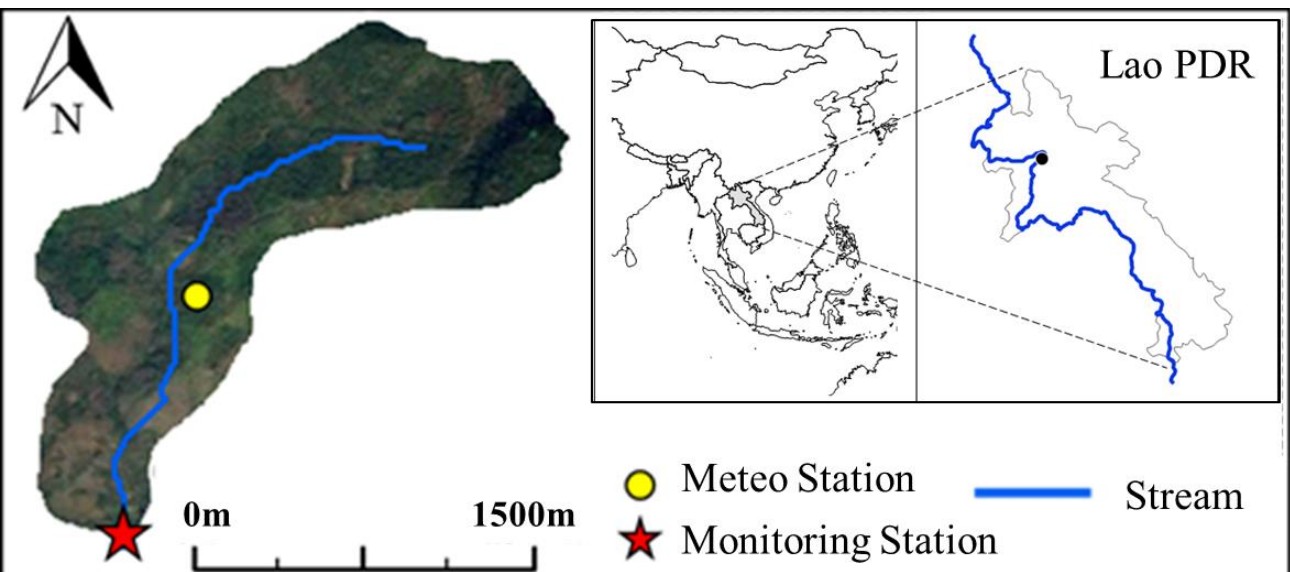


**Figure 1:** Location of the study area. The study area is located near Luang Prabang in northern

Lao PDR. The monitoring station is located at the outlet of the catchment, where water level is

recorded, and where water samples are collected for *E. coli* concentration measurement. Climate

data was measured at the meteorological station.



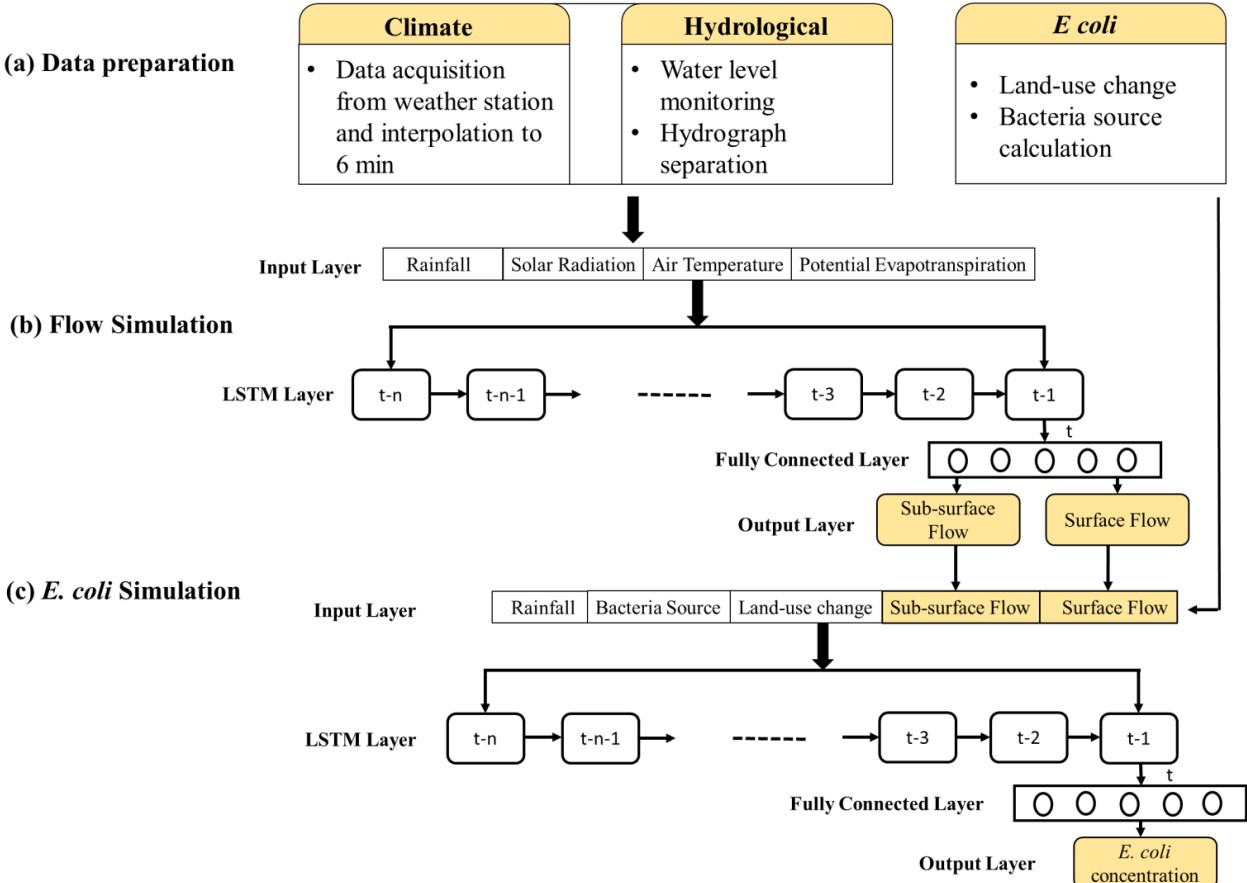



**Figure 2:** Structure of the LSTM Model. Environmental data is used to predict surface flow and
sub-surface flow. Simulated flows along with bacteria source, land-use information and rainfall
is used to simulate *E. coli* concentration. The 'n' represents the length of input data used by
LSTM.



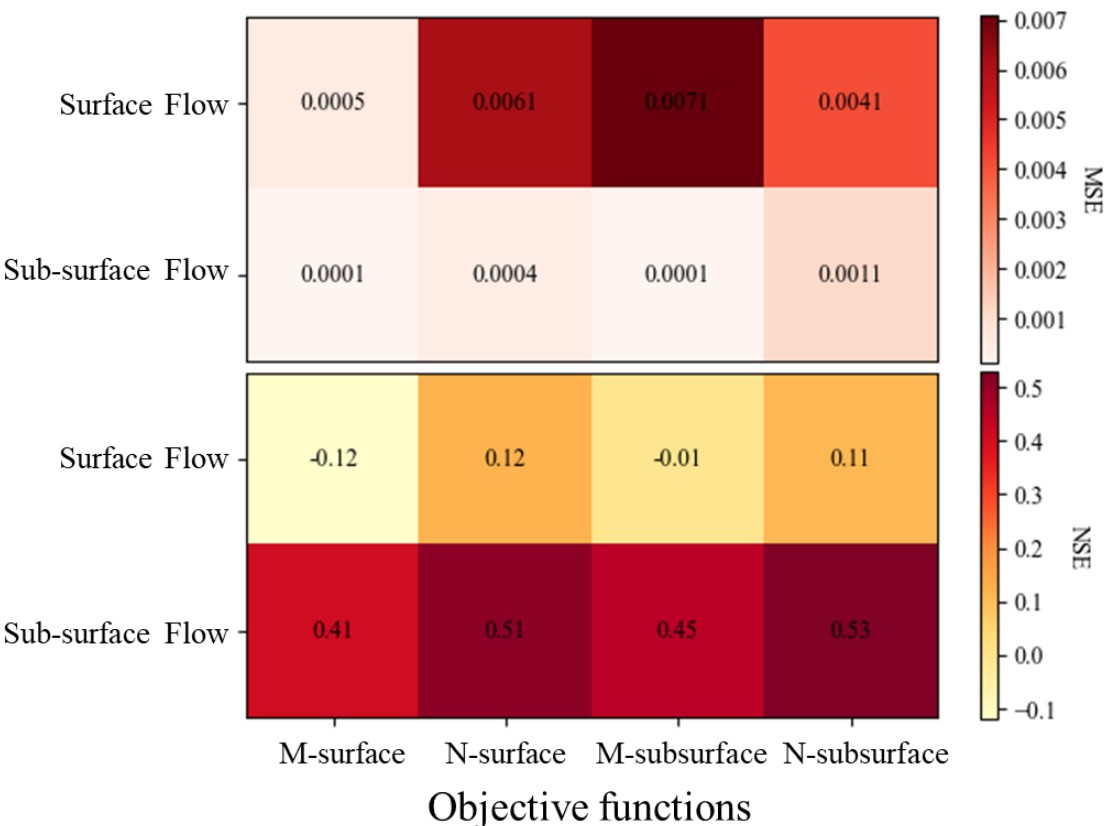




**Figure 3:** Performance of HSPF model with different objective functions (e.g., M-surface, N-Surface, M-subsurface and N-subsurface). The color indicates the value of MSE and NSE. M-surface is the objective function based on MSE and surface flow, N-surface is the objective function based on NSE and surface flow, M-subsurface is the objective function based on MSE and sub-surface flow, and N-subsurface is the objective function based on NSE and sub-surface flow.



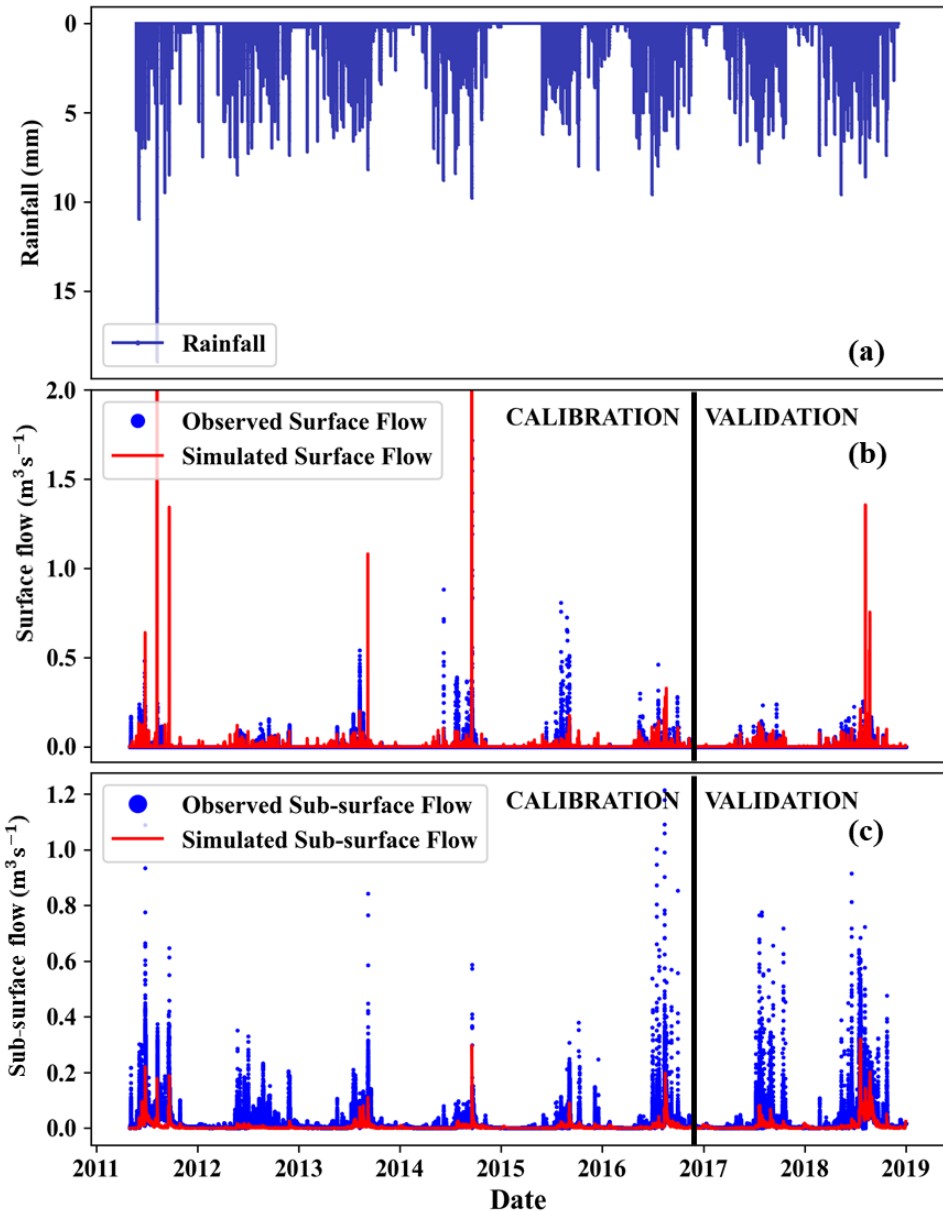


**Figure 4:** Hydrological simulation from HSPF: (a) Measured rainfall, (b) Simulated and

observed surface flow, and (c) Simulated and observed sub-surface flow.



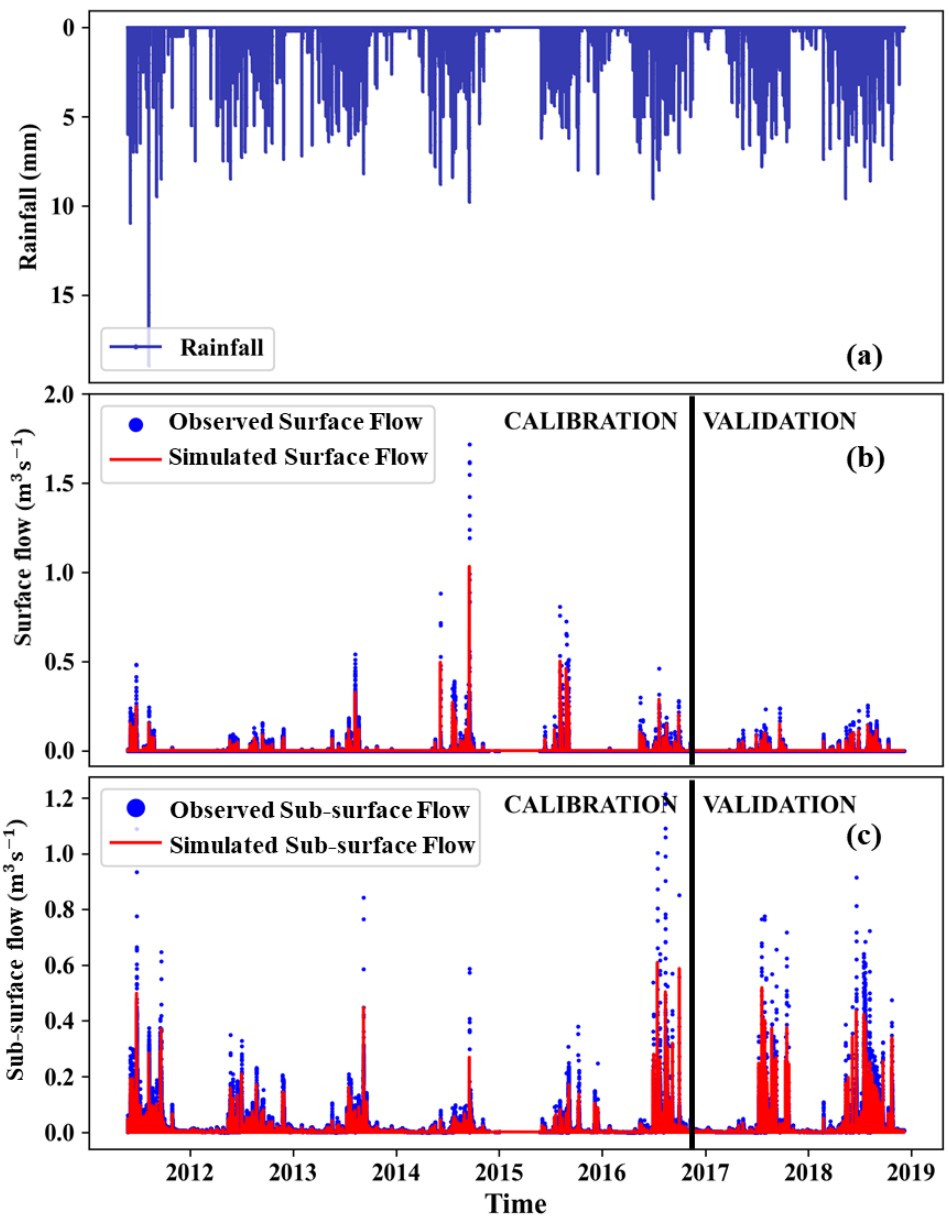


**Figure 5:** Hydrological simulation from LSTM: (a) Measured rainfall, (b) Simulated and

observed surface flow, and (c) Simulated and observed sub-surface flow.




**Figure 6:** Comparison of the hydrological simulation during storm events: (a) MSE value of the surface flow, (b) MSE value of the sub-surface flow, (c) NSE value of the surface flow and, (d) NSE value of the sub-surface flow.




**Figure 7:** *E. coli* simulation from LSTM and HSPF: (a) Measured rainfall, (b) Observed surface

and sub-surface flow, (c) Simulated and observed *E. coli* using HSPF, and (d) Simulated and

observed *E. coli* using LSTM.



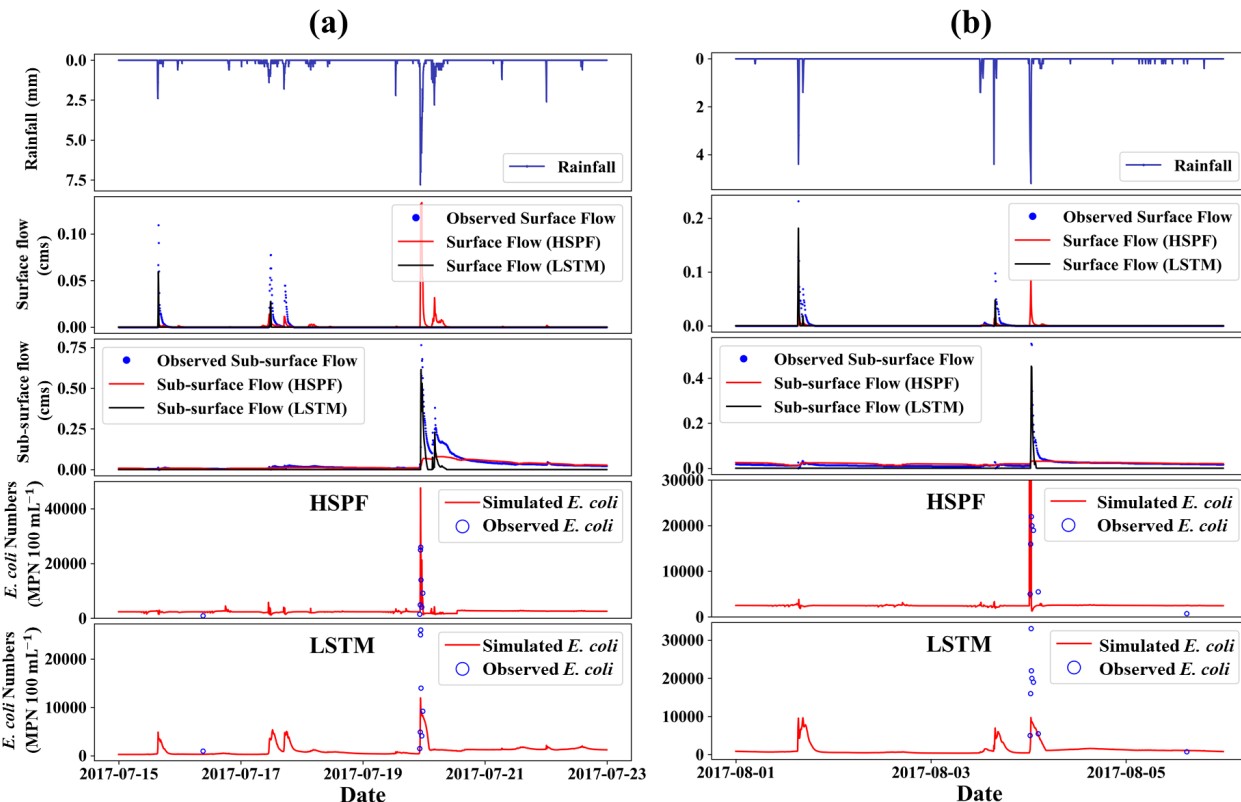

**Figure 8:** *E. coli* concentration of HSPF and LSTM on July 15 and August 1, 2017. Both storm

events were affiliated in validation period.





a)            b)

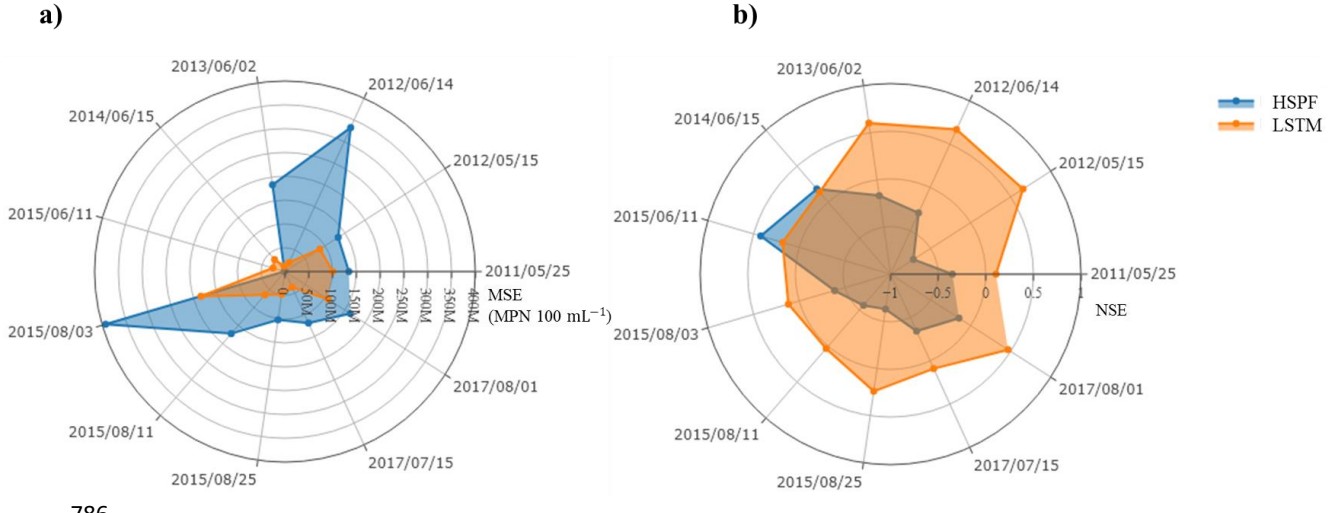

786

**Figure 9:** Comparison of the *E. coli* simulation during storm events: (a) MSE values and (b)

NSE values.



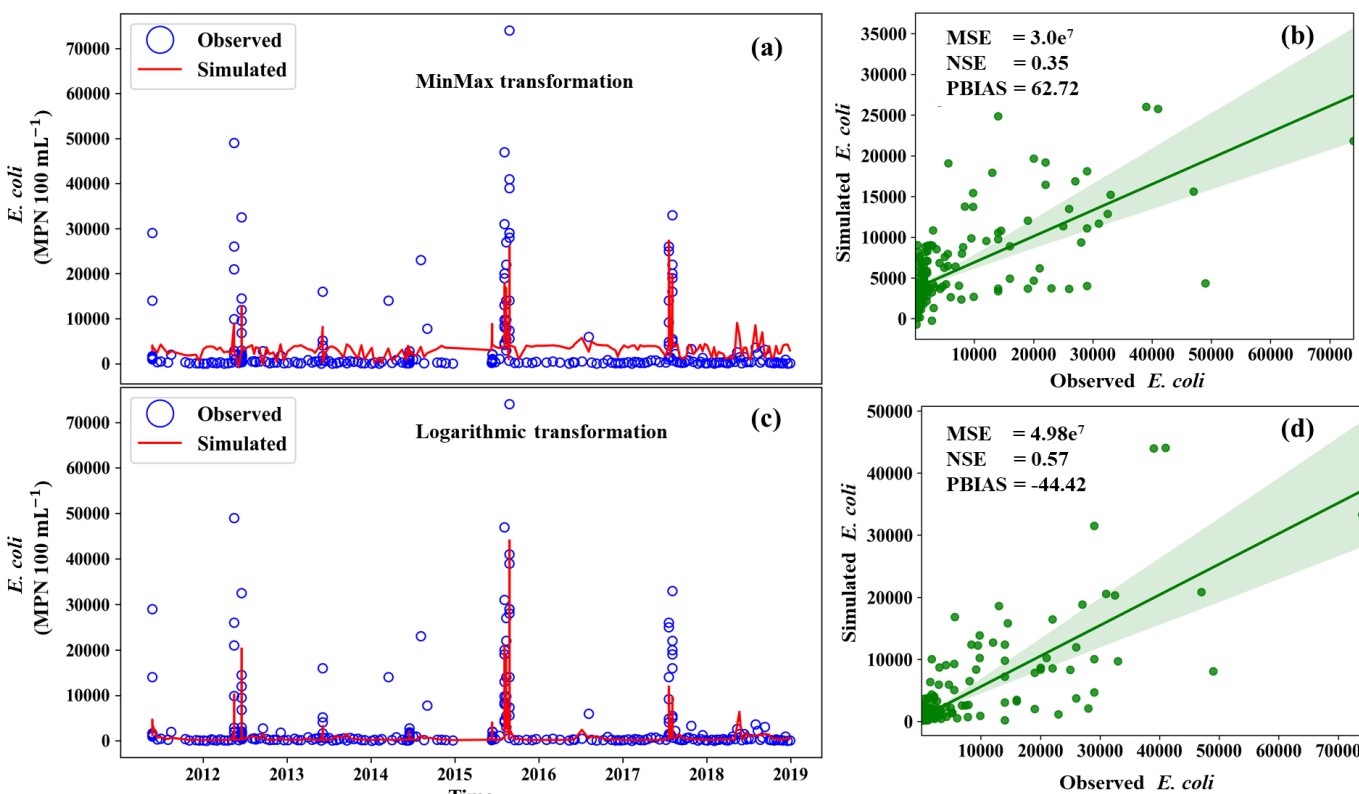

**Figure 10:** Comparison of *E. coil* concentration simulation with the transformation method: (a)

and (c) indicate the *E. coli* simulation using minmax transformation and logarithmic

transformation, respectively. (b) and (d) indicate the scatter plot of *E. coli* using minmax

transformation and logarithmic transformation, respectively.



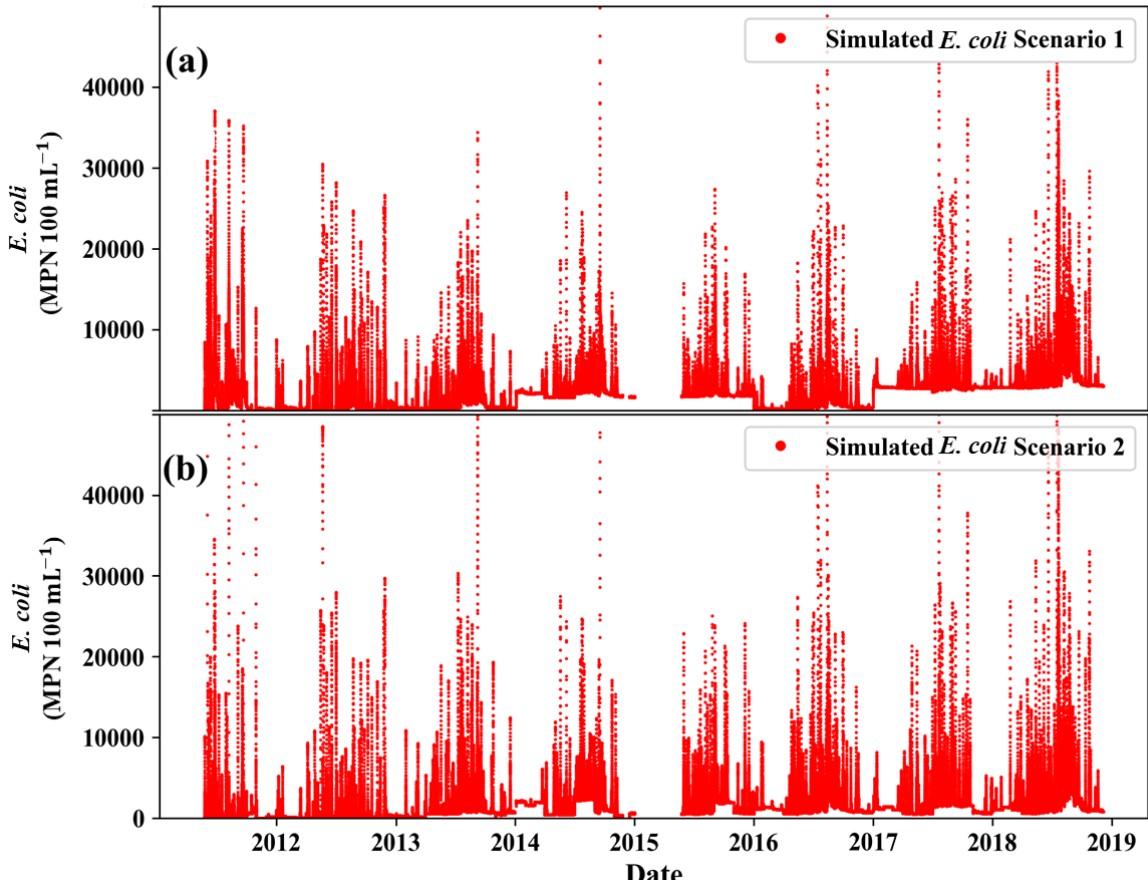

793

**Figure 11:** Changes in *E. coli* sources with land use change scenarios. Scenario 1 used land use

change and bacterial source information. Scenarios 2 used the bacterial source by the fraction of

each land use.