# Peer review of "In-stream Escherichia Coli Modeling Using"

_Hydrology and Earth System Sciences, 2021_

## Author Comment (AC1)

**Reviewer 1**

The authors performed modelling of the transport of Escherichia coli (E. coli) in a tropical headwater catchment located in Lao PDR using a deep learning model and the Hydrological Simulation Program–FORTRAN (HSPF). The deep learning model was built using the long short-term memory (LSTM) technique, whereas the process-based model was constructed using the HSPF. Their results show that the LSTM provided accurate results for surface and subsurface flow, by showing 0.51 and 0.64 of Nash–Sutcliffe Efficiency (NSE), respectively, whereas the NSE values yielded by the HSPF were -0.7 and0.59 for surface and subsurface flow. The simulated E. coli concentration from LSTM also improved, yielding an NSE of 0.35, whereas the HSPF showed an unacceptable performance, with an NSE value of -3.01. The subject is interesting, important and useful. However, there are still some key points need to be addressed. This reviewer recommends to do some revision taking into account the below comments.

We are thankful to the reviewer for his valuable and insightful comments. We have revised the manuscript according to comments of the reviewer. The detailed response to each comment of the reviewer is given below.

We would like to inform the reviewer that the line numbers in our responses correspond to updated manuscript.

Line 62, add g in "determining".

**Response:** We have removed the typo in line 62 and modified the sentence.

**Line 61 – 64:** "Thus, they can be a means to determine the fate and transport of fecal pathogenic microorganisms at the catchment scale by simulating *E. coli* in environmental compartments, such as the soil surface and streams (Ligaray et al., 2016; Pachepsky and Shelton, 2011)".

Line65-84, although the authors have performed a good review of literature of process based models, some latest literatures of water quality should be introduced, such as E.coli (Ligaray et al., 2016; Sowah et al., 2020), and limitations of process-based models (Wang et al., 2020).

https://www.sciencedirect.com/science/article/pii/S0048969720341917?casa_token=itdrAAzone8AAAAA:boSZBTjS_8FQY3RZ2bJz9zwQjiwpz9QOLuLvqK2iB6_CMu7NFaqmjvYcaCwPxwvLO7yw1XEQKA

https://www.sciencedirect.com/science/article/pii/S002216941931248X

https://www.sciencedirect.com/science/article/pii/S0048969720326097

https://www.mdpi.com/2073-4441/13/4/518

**Response:** We thank the reviewer for the literature suggestion. This literature provides important insights about the latest research on *E. coli*. We have included the important highlights from this literature in our manuscript.

**Lines 75 - 81:** Recently, Sowah et al. (2020) applied the SWAT model to research the sources and drivers of *E. coli* in Clouds Creeks watershed, USA. However, the process-based models still have limitations to accuracy due to the complexity of relationships among hydrological and environmental variables (Abimbola et al., 2020). In addition, the simplified equations of these models can increase the inherent uncertainties, resulting in simulation errors. To overcome these limitations, several modifications of the *E. coli* module of the SWAT model have been proposed to incorporate the impacts of the multiple drivers of *E. coli* fate and transport (Cho et al., 2016; Jeon et al., 2019; Meshesha et al., 2020*).*

Line85-95, it is unclear what advantages DL has over process-based models. Are there

any disadvantages of DL, compared to process-based models?

**Response:** Deep learning-based models have the advantage over their process-based counterparts

because of their high accuracy, faster prediction time and their ability to model complex

relationships between input and output features. This ability of deep learning comes mainly from

their ability to exploit special form of compositionality in input data by creating abstract features

in different layers of neural networks. We have added the following lines to explain more clearly

the advantages of deep learning-based models over process-based models.

**Line 88 – 92:** "Deep learning-based models are superior to their process-based counterparts due

to their high accuracy, faster prediction time, and their ability to model complex physical

phenomenon (Sze et al., 2017). Deep learning models can exploit a particular compositionality in

the input features by finding more abstract features in them (Bengio et al., 2021)".

One the other hand, one disadvantage of deep learning models is their lack of explainability i.e. it

is difficult to explain their output. In fact, this is an open research problem and many solutions are

being proposed to answer this question. We have added this point in section 3.6 (Limitations and

future research) of the manuscript.

**Lines 427 – 434:** "The deep learning-based approach can yield high model performance but it has

the limitation in terms of explainability and interpretability (Molnar, 2020). The neural networks

are generally considered as black-box and the question of interpreting them is still an open research

problem (Mitchell, 2021; Tiddi, 2020). Several methods have been proposed to interpret the

behavior of neural networks (Molnar et al., 2020). Explaining the output of neural networks can

enhance the confidence of decision makers (Lipton, 2018). Therefore, we propose future research involving deep learning models will benefit if the questions of interpretability and explainability are considered along with model's prediction performance".

Line376-378, There are no Figs. S2a-S2c in Figure S2.

**Response:** This was mistake to refer Figure S2 here. The actual figure is S1. We have corrected this mistake.

**Lines 399 – 400:** "In scenario 1, we used land-use change time-series information (Fig. S1a) and bacterial source information (Fig. S1b)".

In Fig. S11d and S11e, a peak on 2015-08-28 was captured by both models. However,

It look no data of E. Coli. This should give some explanation if no observed data.

**Response:** We agree with the reviewer that the peak shown by both HSPF and LSTM in Fig. S11 on August 28, 2015. The observed missing *E. coli* concentration during this peak is likely due to lack of observation. We have mentioned this in the manuscript.

**Line 371 – 377:** "We observed that both HSPF and LSTM simulated peaks even when the observed data did show corresponding peaks (Figures S8 and S11). The peaks predicted in Fig. S8 are solely from HSPF while the peak event in Fig. S11 is predicted by both HSPF and LSTM models. This shows the efficacy of both calibrated models. We could conclude from Fig. S11 that the lack of observed peak is more likely because of missing observation. However, a similar conclusion cannot be drawn for all the predicted *E. coli* peaks in Fig S8 because of contradicting results of LSTM and HSPF".

It is unclear where input data sources are from for both LSTM and HSPF. Furthermore, is land use resolution same for both LSTM and HSPF?

**Response:** The input data for both HSPF and LSTM consists of climate, hydrological, E. coli source, and electrical conductivity data. The climate data was measured with an automatic weather station located at the study site. This has been described in following lines in manuscript.

**Lines 120 - 122:** "Rainfall, relative humidity, solar radiation, wind speed, and air temperature were measured with an automatic weather station Campbell Scientific BWS200, which was equipped with ARG100 (a 0.2 mm capacity tipping bucket)".

The electrical conductivity data was used to calculate surface and sub-surface flow. This method has been used in several previous studies such as Ribolzi et al., 2018. We have given the detailed description of this method in supplementary information (Text S2).

**Lines 123 – 128:** "We measured the stream water level at the monitoring station using a V-notch and water-level recorder (OTT Thalimedes). The discharge was estimated based on the rating curve relating discharge to water levels. The surface and subsurface flow were calculated using the electrical conductivity method (Ribolzi et al., 2018). A detailed description of this method is provided in the supplementary information (Text S2)".

The *E. coli* concentration was measured based on the standardized microplate method (ISO 9308-3). The detailed description of the experiments is given in supplementary information provided with the manuscript.

**Lines 128 – 134:** "*E. coli* concentration was measured based on the standardized microplate method (ISO 9308–3). A detailed explanation of *the E. coli* experiment can be found in the supplementary information (Text S3). In this study, we carried out biweekly grab sampling of *E. coli* from 2011 to 2018. Over the same period, we also monitored 11 flood events to assess *E. coli* dynamics during flood events using an automated sampler (ICRISAT) triggered by the water level recorder to collect water after every 2 cm water level change during flood rising and every 5 cm water level change during flood recession".

Fig. 4 and Fig. 5 can be merged to remove a rainfall figure.

**Response:** We have merged Fig. 5 and Fig. 5 into one Fig. 4. Fig. 4 now shows surface and sub-surface flow from HSPF and LSTM models. The subsequent numbering of all the figures in the manuscript has been updated.

[Figure]

**Figure 4:** Hydrological simulation from HSPF and LSTM: (a) Simulated and observed surface flow from HSPF, (b) Simulated and observed sub-surface flow from HSPF, (c) Simulated and observed surface flow from LSTM, and (d) Simulated and observed sub-surface flow from LSTM.

Line 182, it should be briefly described how the data has been converted to a 6 min frequency.

**Response:** The rainfall data was recorded at 6-minute interval for 2011 and 2012. It was recorded with 1-minute frequency from 2013 to 2018. We used cumulated sum of rainfall data from 2013 to 2018 to convert it into 6-minute time-step. Using the automatic weather station, we also recorded hourly relative humidity, solar radiation wind speed and air temperature. These data were then used to calculate potential evapotranspiration using the method of Penman-Monteith at a 1-h time-step. Finally, we interpolated the potential-calculated evapotranspiration to a 6-min time-step. For E. coli, we considered the values nearest to 6-minute time-step as representative of that time-step. We have briefly discussed this in following lines in the manuscript.

**Lines 186-190:** "This was carried by interpolating the hourly weather data. Rainfall data were already available at 6 min for 2011 and 2012 while for 2013 to 2018 it was available at 1 min frequency and was aggregated into a 6-min time series. For E. coli concentration, the values nearest to a 6 min step were used as representative of that time step".

Figs. S6-S11 should be explained and discussed.

**Response:** We have added discussion about hydrological results of HSPF and LSTM which are illustrated in Fig. S6-S11 in the manuscript.

**Line 322 – 327:** "The simulated surface flow by HSPF followed the rainfall events more closely as compared to that of LSTM. The peaks in surface flow in Fig. S8 are completely missed by LSTM while captured by HSPF model. We also observed that LSTM can follow the observed trends in surface and subsurface flow more closely than the HSPF (Fig. S6, S9, S10). The falling limb from the predicted sub-surface flow of LSTM is gentle and follows the observed pattern (Fig. S9 - S11)".

A discussion paragraph about results of E. coli in Fig. S6 – S11 have been added in response to reviewer's previous comment.

**Line 371 – 377:** "We observed that both HSPF and LSTM simulated peaks even when the observed data did show corresponding peaks (Figures S8 and S11). The peaks predicted in Fig. S8 are solely from HSPF while the peak event in Fig. S11 is predicted by both HSPF and LSTM models. This shows the efficacy of both calibrated models. We could conclude from Fig. S11 that the lack of observed peak is more likely because of missing observation. However, a similar conclusion can not be drawn for all the predicted *E. coli* peaks in Fig S8 because of contradicting results of LSTM and HSPF".

**References**

Abimbola, O. P., Mittelstet, A. R., Messer, T. L., Berry, E. D., Bartelt-Hunt, S. L., and Hansen, S. P.: Predicting Escherichia coli loads in cascading dams with machine learning: An integration

of hydrometeorology, animal density and grazing pattern, Science of The Total Environment, 722, 137894, 2020.

Bengio, Y., Lecun, Y., and Hinton, G.: Deep learning for AI, Communications of the ACM, 64, 58-65, 2021.

Cho, K. H., Pachepsky, Y. A., Kim, M., Pyo, J., Park, M.-H., Kim, Y. M., Kim, J.-W., and Kim, J. H.: Modeling seasonal variability of fecal coliform in natural surface waters using the modified SWAT, Journal of Hydrology, 535, 377-385, 2016.

Jeon, D. J., Ligaray, M., Kim, M., Kim, G., Lee, G., Pachepsky, Y. A., Cha, D.-H., and Cho, K. H.: Evaluating the influence of climate change on the fate and transport of fecal coliform bacteria using the modified SWAT model, Science of the Total Environment, 658, 753-762, 2019.

Ligaray, M., Baek, S. S., Kwon, H.-O., Choi, S.-D., and Cho, K. H.: Watershed-scale modeling on the fate and transport of polycyclic aromatic hydrocarbons (PAHs), Journal of hazardous materials, 320, 442-457, 2016.

Lipton, Z. C.: The Mythos of Model Interpretability: In machine learning, the concept of interpretability is both important and slippery, Queue, 16, 31-57, 2018.

Meshesha, T. W., Wang, J., and Melaku, N. D.: A modified hydrological model for assessing effect of pH on fate and transport of Escherichia coli in the Athabasca River basin, Journal of Hydrology, 582, 124513, 2020.

Mitchell, M.: Why AI is harder than we think, arXiv preprint arXiv:2104.12871, 2021.

Molnar, C.: Interpretable machine learning, Lulu. com2020.

Molnar, C., Casalicchio, G., and Bischl, B.: Interpretable machine learning–a brief history, state-of-the-art and challenges, Joint European Conference on Machine Learning and Knowledge Discovery in Databases, 417-431,

Pachepsky, Y. and Shelton, D.: Escherichia coli and fecal coliforms in freshwater and estuarine sediments, Critical reviews in environmental science and technology, 41, 1067-1110, 2011.

Ribolzi, O., Lacombe, G., Pierret, A., Robain, H., Sounyafong, P., De Rouw, A., Soulileuth, B., Mouche, E., Huon, S., and Silvera, N.: Interacting land use and soil surface dynamics control groundwater outflow in a montane catchment of the lower Mekong basin, Agriculture, Ecosystems & Environment, 268, 90-102, 2018.

Sowah, R. A., Bradshaw, K., Snyder, B., Spidle, D., and Molina, M.: Evaluation of the soil and water assessment tool (SWAT) for simulating E. coli concentrations at the watershed-scale, Science of the Total Environment, 746, 140669, 2020.

Sze, V., Chen, Y.-H., Yang, T.-J., and Emer, J. S.: Efficient processing of deep neural networks: A tutorial and survey, Proceedings of the IEEE, 105, 2295-2329, 2017.

Tiddi, I.: Directions for explainable knowledge-enabled systems, Knowledge Graphs for eXplainable Artificial Intelligence: Foundations, Applications and Challenges, 47, 245, 2020.

Wang, L., Chen, J., and Marathe, M.: Tdefsi: Theory-guided deep learning-based epidemic forecasting with synthetic information, ACM Transactions on Spatial Algorithms and Systems (TSAS), 6, 1-39, 2020.

---

## Author Comment (AC2)

**Reviewer 2**

It is important and meaningful to improve prediction accuracy in modeling works. Nowadays, application of machine learning including deep learning techniques may be very promising to support conventional modeling approaches. This study constructed LSTM model to simulate surface/subsurface flow and E. coli concentration in a catchment and compared the performance with HSPF model, which is a well-known watershed model. The results are quite interesting and can be useful in scientific and practical fields. I think that this study can be considered as a publication in the journal with **minor revision**.

Some comments are as follows.

We thank the reviewer for finding our work interesting and useful. We have revised the manuscript with reviewer's comments. The answer to each of the comments are given below.

In construction of LSTM in a catchment, this study used only meteorological data as an input to predict flow rates. An issue is that how we can consider characteristics of catchment such as land use and soil property in simulation of flow rate.

**Response:** We agree with the reviewer that the flow rates in a catchment are affected by the catchment characteristics such as soil characteristics, slope etc. The LSTM is in principle designed to extract temporal features from time-varying input data. The static data consisting of catchment characteristics can however be fed to the along with continuous data. However, in this study we did not consider this because the study consisted of only single catchment. In such a scenario, the LSTM will be trained with only single constant value for each catchment feature. On the other hand, if had data for several catchments, then LSTM could be trained to learn different catchment characteristics. We adopted similar strategy in the preceding study (Abbas et al., 2020) where we

trained LSTM with input data of different HRUs and the static input data of HRUs was used along with time series data of each HRU.

I think that the basin area in this study may be relatively small. What if LSTM model application in a largescale watershed? Is the meteorological data enough to predict flow rate? I think that this discussion can be very informative to readers in LSTM application to watershed scale.

**Response:** The prediction performance of deep learning models is strongly affected by the data distribution of the training data. In order to make a regional or global prediction model, the LSTM should also be trained with input data from more catchments. Some recent work is being carried out in this direction to build regional streamflow prediction models using deep learning. However, due to scarcity of water quality data, building such a regional model is more challenging. Nevertheless, we think the approach adopted in this study can be used as guideline for building regional water quality model by training the model with more input data. We have added this discussion in the manuscript.

**Lines 432 – 441:** "Deep learning models are based upon the (independent and identically distributed) (IID) assumption which means that the validation data is expected to have the same distribution as that of the training data (Kawaguchi et al., 2017). However, this is not a realistic assumption and it is considered as one of the challenges for researchers in machine learning (Bengio et al., 2021). Thus, in order to build regional or global hydrological models, the deep learning model should be trained on catchment data from diverse catchments. Several researchers have adopted this approach to build regional models for streamflow prediction (Anderson and Radic, 2021; Kratzert et al., 2019; Xiang et al., 2021). However, a similar approach for building regional water quality model will be more challenging due to the scarcity of water quality data.

We hope that the lessons from this study can be used a guideline to train neural networks on regional water quality data".

Line 29, full name is needed for "PDR"

**Response:** We have corrected this by writing the full name of PDR. The modified sentence is as follow

**Line 29:** In this study, we simulated the fate and transport of *Escherichia coli* (*E. coli*) in a 0.6 km² tropical headwater catchment located in Lao People's Democratic Republic (Lao PDR) using a deep learning model and a process-based model.

Line 52, what is the meaning of "less dangerous than other pathogens"?

**Response:** We agree with the reviewer that the word 'less dangerous' is vague and not clear. Therefore, we have removed this word from the sentence. The modified sentence in manuscript is as follows;

**Line 52 – 53:** "*Escherichia coli* (*E. coli*) has been frequently used as an indicator of fecal bacteria because it is easy to culture (Rochelle-Newall et al., 2015)".

In study site description, basin area is needed.

**Response:** We have added the catchment area in the study site description.

**Line 112:** "The study area is 0.6 km$^2$ the Houay Pano headwater catchment".

Line 167, what is the meaning of "rewrote" Did you modify the source code? Rephrase it.

**Response:** The original HSPF code is in FORTRAN programming language which is difficult to use with modern optimization algorithms and change. Therefore, we converted the code into Python programming language. Thus, we did not modify the source code but converted it into Python programming language.

**Lines 170 – 171:** For this study, we converted the original FORTRAN code of *E. coli* module of HSPF into Python programming language.

Line 192-193 and 376-378, It is difficult to understand scenario 1 and scenario 2. Scenario 1 is land use change with same E. coli loading (Fig. S1-a and b) and scenario 2 is land use change with variable E. coli loading in terms of land use (Fig. S1-a and c))? It is confusing.

**Response:** The purpose of these two scenarios is to assess the impact of different input features. In scenario 1, the land use change information and *E. coli* source information is represented by separate input features. This results in increase in number of input features. In scenario 2, the number of input features was reduced by combining the land-use change information with that of *E. coli* concentration. We have rephrased the sentences to make it clearer.

**Line 205 – 208:** "In scenario 2, the number of input features was reduced by multiplying *E. coli* source with land-use change. In this way, we calculated *E. coli* source per area for each land use and used this as input instead of using land use and E. coli information as separate input features".

Line 265, among the 10 most sensitive parameters? 10 variables are equally sensitive?

**Response:** The sensitivity of the 10 parameters is not equal. The sensitivity rank of these parameters is given in Table S2.

Table 2 and line 313, number of optimal batch size and lookback steps are mismatched between the table and sentence. 128 vs 100 and 50 vs 5 h

**Response:** We thank the reviewer for pointing the mistake. We have corrected the values of batch size and hidden units for LSTM for surface and sub-surface flow estimation. However, the value of lookback steps is correct i.e. 5 hours. This is because the 5 hours of historical data was used as input for both flow estimation E. coli concentration estimation.

**Lines 331 – 332:** The optimal batch size and LSTM units were 128 and 64, respectively.

Figure 8, what is (a) and (b) in the figure? (a) July 15 and (b) August 1?

**Response:** In Fig. 8 (a) and (b) represents prediction performance of models during two selected periods with several storm events. The first period is from July 15 to July 23 while the second period is from August 1 to August 5 2017. We have rephrased the caption of Fig. 8 to make this clearer.

**Figure 8:** *E. coli* concentration of HSPF and LSTM during July 15-22 (a) and August 1-5, 2017 (b). Both storm events were affiliated in validation period.

Line 362 – 370, where is minmax and logarithmic transformation from? There was not any mention about application of minmax and logarithmic transformation in method. All E. coli simulations was based on logarithmic?

**Response:** We have added information about minmax and logarithmic transformation of the E. coli data in the methods section (2.2.2). We trained the neural network on the transformed data. However, we calculated the performance metrics by transforming the predictions back to normal scale.

**Line 198 – 202:** "The preprocessing of the data before feeding the neural network can have a significant impact on the of performance (Banhatti and Deka, 2016). Therefore, we compared the performance of model by transforming the *E. coli* concentration using the minmax transformation and the logarithmic transformation. The minmax transformation results in data between 0 and 1 while logarithmic transformation transforms the data on a logarithmic scale".

Line 376-378, Referencing of the figure in the sentences may be wrong. Not Fig. S2 but Fig. S1. Check it..

**Response:** We are thankful to the reviewer for pointing out this mistake. We have corrected this mistake.

**Lines 399 – 400:** In scenario 1, we used land-use change time-series information (Fig. S1a) and bacterial source information (Fig. S1b).

**References**

Abbas, A., Baek, S., Kim, M., Ligaray, M., Ribolzi, O., Silvera, N., Min, J.-H., Boithias, L., and Cho, K. H.: Surface and sub-surface flow estimation at high temporal resolution using deep neural networks, Journal of Hydrology, 590, 125370, 2020.

Anderson, S. and Radic, V.: Evaluation and interpretation of convolutional-recurrent networks for regional hydrological modelling, Hydrology and Earth System Sciences Discussions, 1-43, 2021.

Banhatti, A. G. and Deka, P. C.: Effects of Data Pre-processing on the Prediction Accuracy of Artificial Neural Network Model in Hydrological Time Series, in: Urban Hydrology, Watershed Management and Socio-Economic Aspects, Springer, 265-275, 2016.

Bengio, Y., Lecun, Y., and Hinton, G.: Deep learning for AI, Communications of the ACM, 64, 58-65, 2021.

Kawaguchi, K., Kaelbling, L. P., and Bengio, Y.: Generalization in deep learning, arXiv preprint arXiv:1710.05468, 2017.

Kratzert, F., Klotz, D., Shalev, G., Klambauer, G., Hochreiter, S., and Nearing, G.: Towards learning universal, regional, and local hydrological behaviors via machine learning applied to large-sample datasets, Hydrology and Earth System Sciences, 23, 5089-5110, 2019.

Rochelle-Newall, E., Nguyen, T. M. H., Le, T. P. Q., Sengtaheuanghoung, O., and Ribolzi, O.: A short review of fecal indicator bacteria in tropical aquatic ecosystems: knowledge gaps and future directions, Frontiers in microbiology, 6, 308, 2015.

Xiang, Z., Demir, I., Mantilla, R., and Krajewski, W. F.: A Regional Semi-Distributed Streamflow Model Using Deep Learning, 2021.